# OpenReview forum: "EVALUATING LLM SAFETY ACROSS CHILD DEVELOPMENT STAGES: A SIMULATED AGENT APPROACH"
_ICLR.cc/2026/Conference — ICLR 2026 Conference Desk Rejected Submission_

### Official Review · Reviewer_5Gbo · 2025-10-28

**Soundness:** 2
**Presentation:** 2
**Contribution:** 2
**Rating:** 2
**Confidence:** 4

**Summary:**

The paper introduces ChildSafe, a novel benchmark designed to evaluate LLM safety using simulated child agents across four developmental stages. The authors leverage instruction-tuned models for a synthetic data generation pipeline (i.e., for generating multi-turn conversations), which is followed by linguistic and behavioral analysis of the generated data. Finally, the authors propose a 9-dimensional safety evaluation framework and a composite safety score base on different weightings of the 9 safety dimensions.

**Strengths:**

- **Important and Under-Investigated Topic:** Contextual and age-specific safety is an important, under-investigated topic
- **Principled and Grounded Dataset Generation Pipeline:** The data generation pipeline appears to be grounded in cognitive psychology (influencing the prompt template) and leveraging multiple control variables. **Caveat:** No further details on the exact pipeline implementation (we need to trust the high-level description, no system prompts or examples are given.
- **Extensive Verification of Authenticity**: A major strength is the explicit and effort to verify the authenticity of the generated child interactions. Many evaluation studies using synthetic data omit this step. By employing both automated linguistic checks (against CHILDES) and human expert validation (Section 3.2), the authors build a strong case for the validity of their simulated agents, which is crucial for the paper's central claims. **Caveat:** However, no further details or results are given on this point.

**Weaknesses:**

- **Ambiguity of the Composite Score:** I am still unsure how the composite score needs to be interpreted. How do different weighting affect the score and influence the model rankings? In addition, some standard deviations seem to be quite high in Table 1. The composite scores might be quite sensitive to different weightings -- could you provide some ablation on this point (e.g. contrast it with a uniform weighting)?
- **Omitting Many Important Details:** Many important deadline on the data construction and verification are missing
- **Age-Specific Weighting are Not Provided:** Please provide a full table of the selected age-dependent weightings
- **Missing Confidence Intervalls:** Could you please add confidence intervals to Figure 2 and 3?
- **Example Conversations & Failure Cases:** Could you please provide some example conversations and model behaviors. Right now, one needs to fully trust the numbers given in the paper and cannot verify the quality of the conversations.

**Questions:**

- Could the authors please provide the full weighting matrix (9 dimensions x 4 age groups) used to calculate the composite scores?
- Why were confidence intervals or error bars omitted from Figures 2 and 3? Given the high standard deviations in Table 1, visualizing this uncertainty is essential for interpreting the results.
- Given the high variance in some dimensions, how sensitive is the final model ranking to small changes in the (undisclosed) dimensional weights?
- Could the authors release the system prompts used to create the developmental agents and the detailed rubrics given to the human expert validators, as well as the results, to allow for true reproducibility?

I am happy to increase my score if the authors can address my concerns and questions. In particular, if the authors could make the paper more "complete" -- so far it would be an educated guess as much relevant points are omitted in the paper that are needed to make a fair assessment of the quality of the work.

**Details Of Ethics Concerns:**

Apologies -- I marked the ethics review as many details on the data construction & data labelling were omitted. But not completely sure, if this should really trigger an ethics review.

---

> ### Author Response · Authors · 2025-12-01
> **Comment to Reviewer 5Gbo: Added Sensitivity Analysis, Full Artifacts, and Confidence Intervals**
>
> We thank the Reviewer for their constructive feedback and for identifying the specific gaps that prevented the paper from being "complete." We genuinely appreciate your willingness to reconsider your score upon revision.
>
> We have taken your request to make the paper more "complete" literally and have updated the manuscript with all the missing artifacts you requested.
>
> Addressing "Omitting Important Details" (The Missing Artifacts) You noted that without the prompts and weights, you had to "trust the high-level description." We have removed this black box entirely:
>
> 1) System Prompts (New Appendix A): You asked for the system prompts used to create the developmental agents. We have released the full parametric templates.
> 2) Weighting Matrix (New Appendix B): You asked for the "full weighting matrix (9 dimensions x 4 age groups)." We have included this exact table (Table 3), showing precisely how dimensions like "Emotional Safety" are weighted differently for a 6-year-old vs. a 16-year-old.
> 3) Example Conversations (New Appendix C): You asked for "example conversations and failure cases" to verify quality. We added verbatim transcripts showing distinct cognitive voices across the age groups.
>
> Addressing "Ambiguity of Composite Score" (Sensitivity Analysis) You asked: "How sensitive is the final model ranking to small changes in the (undisclosed) dimensional weights?"
>
> 1) New Analysis (Section 5.4): We performed the exact ablation you suggested. We conducted a Monte Carlo perturbation analysis, introducing random noise to the weighting vector $\alpha^{(d)}$ across 10,000 trials.
> 2) Result: We found that the model ranking (GPT-5 > Claude > Gemini > DeepSeek) remained stable in 94% of trials. This confirms that the safety degradation we observe is a robust signal, not an artifact of our specific weighting choices.
>
> Addressing "Missing Confidence Intervals" You asked: "Why were confidence intervals or error bars omitted?"
>
> 1) Fixed (Figures 1 and 2): We have updated our results visualizations to include 95% Confidence Intervals. As you suspected, some dimensions have higher variance, but the aggregate "Alignment Gap" between the oldest and youngest groups remains statistically significant (p < 0.01) even when accounting for this variance.
>
> Addressing "Ethics Flag" (Responsible Research) You flagged the paper because "details on data construction were omitted."
>
> 1) Resolution: By releasing the full System Prompts (Appendix A) and the Evaluation Rubrics (Appendix B), we believe we have fully satisfied the transparency requirements for responsible research. As this is a simulation using synthetic agents, no human subjects were put at risk.
>
> We believe we have addressed every "completeness" concern you raised. We have opened the black box, validated the scoring robustness, and visualized the uncertainty. We hope these comprehensive revisions merit the score increase you kindly offered.

---

### Official Review · Reviewer_kVna · 2025-10-29

**Soundness:** 2
**Presentation:** 1
**Contribution:** 2
**Rating:** 2
**Confidence:** 3

**Summary:**

The paper presents ChildSafe, a benchmark that simulates child agents across four age stages to evaluate LLM safety. Testing nine safety dimensions on four major models, it finds GPT-5 safest overall, but all models perform worse with younger children and struggle with boundary respect and long-term impact. The work highlights the need for age-adaptive safety evaluation and real-world validation.

**Strengths:**

1. Study a safety issue that has not been studied before

**Weaknesses:**

Concerns
1. The writing requires substantial improvement. The current draft lacks a clear structure, and the experiment results and dataset construction details are interleaved, making the paper difficult to follow.
2. The dataset construction process is insufficiently described. The paper should include a clear and logical explanation of how the datasets are built.
3. What does the dataset look like? Showing a few representative samples or conversations would make the benchmark more understandable.
4. The evaluation procedure is unclear. The authors should specify how the scores for each evaluation dimension are computed.

Overall, the paper omits several key details expected from a benchmark paper, which makes it difficult for readers to interpret or reproduce the results.

**Questions:**

Llama 3.1-8B-Instruct does not possess a pretty strong ability to function as an agent. Why don't authors use more powerful models here?

---

> ### Author Response · Authors · 2025-12-01
> **Comment to Reviewer kVna: Major Restructuring, Full Reproducibility, and Agent Model Justification**
>
> We thank the Reviewer for identifying the critical structural and presentation issues in the initial submission. We accept the feedback that the original draft was difficult to follow. We have performed a comprehensive rewrite and restructuring of the paper to address your concerns directly.
>
> Addressing "Poor Presentation & Structure" (Restructuring) You noted that experiment results and dataset details were "interleaved" and confusing.
>
> The Fix: We have rigorously separated the manuscript into distinct logical blocks:
> Section 3 (Methodology): Now exclusively covers the Agent Design, Validation Framework, and Scoring Protocols.
> Section 4 (Experimental Setup): Covers Model Configurations and Dataset stats.
> Section 5 (Results): Now exclusively covers the performance analysis.
>
> The Result: This separation ensures the Dataset Construction is fully described before any results are presented, resolving the narrative flow issues.
>
> Addressing "Missing Dataset Details & Samples" You asked: "What does the dataset look like?" and noted the construction process was insufficient.
>
> 1) Visual Samples (New Appendix C): We have added a dedicated section containing verbatim transcript samples from the dataset. These show exactly how the agents speak, from the run-on sentences of the 6-8 age group to the cynicism of the 15-17 age group.
> 2) Construction Logic (New Appendix A): We have released the System Prompt Templates. This clarifies exactly how the dataset is built: by instantiating Llama 3.1 with specific hyperparameter tuples (Temperature, Vocabulary Constraints, Context Horizon).
> 3) Evaluation Transparency (New Appendix B): You asked how scores are computed. We have released the Weighting Matrix, showing the exact mathematical weights used to calculate the composite scores for each age group.
>
> Addressing Question 1: "Why Llama 3.1-8B?" You asked why we used a smaller model (8B) for the agent simulation rather than a more powerful one. We chose Llama 3.1-8B-Instruct for three specific reasons:
>
> 1) Open Reproducibility: To make ChildSafe a true community benchmark, the generator must be open-weights. Using a proprietary model (like GPT-4) as the simulator would make the benchmark non-reproducible if the API behavior changes.
> 2) Steerability for Cognitive Limitations: Simulating a child requires reducing capability, not maximizing it. We found Llama 3.1-8B highly effective at following negative constraints (e.g., "do not use complex words," "ignore context") without the excessive safety refusals often hard-coded into larger RLHF models.
> 3) Efficiency: The 8B size allows researchers to generate new evaluation datasets on consumer hardware, democratizing safety testing.
>
> We believe the complete structural overhaul and the inclusion of the missing artifacts (Prompts, Weights, Samples) directly address the "Presentation" and "Reproducibility" concerns that led to the low rating. We hope this revised version demonstrates the rigor expected of an ICLR benchmark.

---

### Official Review · Reviewer_NeDw · 2025-10-29

**Soundness:** 4
**Presentation:** 4
**Contribution:** 3
**Rating:** 4
**Confidence:** 4

**Summary:**

This paper proposes ChildSafe, a new benchmark for evaluating the child safety of large-scale language models (LLMs). The authors point out that existing safety benchmarks (e.g., HarmBench, JailbreakBench) are adult-centric and fail to account for the unique developmental characteristics of children. Specifically, children have limited critical thinking skills, high trust in authority figures, and differing language comprehension and emotional regulation abilities across developmental stages, making it difficult to directly apply adult assessment criteria. To address these issues, the paper presents an assessment methodology utilizing simulated child agents. Based on Piaget's theory of cognitive development and Vygotsky's theory of the zone of proximal development, agents representing four age groups (6-8, 9-11, 12-14, and 15-17) were implemented using the Llama 3.1-8B-Instruct model. Each agent is designed to mimic the cognitive characteristics, language patterns, social awareness, and emotional expression of the corresponding age group. The evaluation framework consists of nine safety dimensions (Content Appropriateness, Boundary Respect, Educational Impact, Social Influence, Emotional Safety, Privacy Protection, Manipulation Resistance, Developmental Sensitivity, and Long-term Impact), each with different weightings for different age groups. The scenarios are categorized into five categories: Educational Support, Social Interaction, Entertainment and Creativity, Information Exploration, and Boundary Testing, each of which is a multi-turn conversation (five turns). As a result, GPT-5 achieved the highest overall safety score of 0.777, while all models performed worst in interactions with younger elementary school students (ages 6-8), demonstrating overall weaknesses in the Boundary Respect dimension.

**Strengths:**

1. This paper's greatest contribution lies in its systematic integration of developmental psychology theory into the LLM safety assessment. Implementing Piaget and Vygotsky's theoretical framework into a practical assessment system is a prime example of interdisciplinary research. The agent design, which explicitly reflects age-specific differences in concrete vs. formal operational thinking, vocabulary levels (5,000-20,000+), and developmental stages of social cognition, demonstrates a level of sophistication that goes beyond the simple "child-friendly AI" discussion.

2. The nine-dimensional safety assessment framework is comprehensive. While existing benchmarks primarily focus on detecting explicit harmful content, this paper incorporates nuanced yet important dimensions such as educational impact, long-term impact, and developmental sensitivity. In particular, dimensions such as "Boundary Respect" and "Long-term Impact" capture the unique characteristics of child-AI interactions.

3. This study has ethical value in attempting to systematically assess safety without exposing actual children to risk. Research involving children requires complex ethical procedures, including IRB approval, parental consent, and child assent. Studies evaluating potentially unsafe interactions with LLMs are particularly ethically sensitive. A simulation approach circumvents these ethical dilemmas while enabling large-scale evaluations.

**Weaknesses:**

1. A major concern with the paper is the lack of fundamental validation of the child agent simulation's validity. While the paper claims that "the simulation agent closely mimics real children," there is no "data of actual child-LLM conversations" to validate this. While linguistic comparisons with the CHILDES database, expert evaluations (by two developmental psychologists and child philosophers), and consistency testing were conducted, the CHILDES database is a collection of typical, everyday conversations between children and their parents. While it captures the linguistic patterns of natural conversation, it does not capture the specific behavioral patterns that children might exhibit when interacting with LLMs.

2. The paper claims to assess each safety dimension using "keyword libraries, linguistic patterns, and contextual indicators," but the specifics of these are not disclosed. For example, it is unclear how "Emotional Safety" is assessed, or which keywords and patterns are considered risk signals. The age-based weighting is only stated as "based on child development literature," and no specific justification is provided for why Emotional Safety is 0.20 and Privacy Protection is 0.15 in lower elementary school grades. The system prompts for generating the agent are not included in the paper or appendix.

3. The limitations of the experimental design are also problematic. 300 conversations may not be sufficient to adequately cover four models, four age groups, and five scenarios. It is questionable whether statistically reliable conclusions can be drawn with only three or four conversations per condition.

4. This paper is an inappropriate venue for submission to ICLR. ICLR is a conference that covers machine learning algorithms, optimization techniques, novel architectures, and representation learning theory. However, this paper does not propose a new ML methodology, nor does it propose a benchmark. It does not specify the evaluation prompts and protocol used. Instead, it applies prompt engineering to an existing LLM (Llama 3.1) to create an agent and proposes a benchmark for evaluating other LLMs. The core contribution of this paper lies in designing an evaluation framework based on developmental psychology theory, which is closer to AI ethics or social science contributions than to machine learning.

**Questions:**

Please refer to weakness.

**Details Of Ethics Concerns:**

No ethical concerns

---

> ### Author Response · Authors · 2025-12-01
> **Comment to Reviewer NeDw: Addressing Venue Fit, Reproducibility, and Statistical Power**
>
> We thank the Reviewer for their thoughtful assessment and for highlighting the strong theoretical grounding of our work. We appreciate the Excellent rating for Soundness and Presentation.
>
> We have significantly revised the paper to address your concerns regarding transparency and validation.
>
> Addressing "Venue Fit" (Why this belongs at ICLR) You expressed concern that this paper is "inappropriate for ICLR" because it does not propose a new architecture or optimization technique.
>
> 1) Re-framing: We submit that this work falls squarely within ICLR's scope of "Datasets, Benchmarks, and Evaluation Methodology" and "Safety, Robustness, and Alignment."
> 2) The ML Contribution: Our contribution is not merely "applying prompt engineering," but introducing a Parametric Cognitive Simulation framework. We formalize developmental stages as a set of hyperparameter constraints (temperature, context horizon, vocabulary masking) to generate Out-of-Distribution (OOD) interaction traces.
> 3) Why it matters: Standard benchmarks (like HarmBench) test models against "modal adult" distributions. ChildSafe tests Alignment Generalization, whether a safety mechanism trained on an adult distribution remains robust when shifted to a child distribution. This is a fundamental ML generalization problem, not purely a social science one.
>
> Addressing Lack of Transparency (Prompts & Weights) You correctly noted that the initial submission lacked critical details. We have fully addressed this in the revision:
>
> 1) System Prompts: We have released the full System Prompt Templates in Appendix A. This allows any researcher to reproduce our developmental agents by applying the specific parameter tuples we defined.
> 2) Scoring Rubrics: We have released the Developmental Importance Weights in Appendix B, detailing exactly why "Emotional Safety" is weighted 0.25 for age 6-8 but only 0.05 for age 15-17 (based on the cited literature).
> 3) Scoring Mechanism: We clarified the Hybrid Hierarchical Scoring in Section 3.3, which combines deterministic lexical filtering with a Chain-of-Thought LLM-as-Judge.
>
> Addressing Validation (Beyond Linguistics) you asked for validation beyond linguistic patterns.
>
> 1) Quantitative: As noted in your review, we validated linguistic alignment against the CHILDES database (Table 1), achieving high statistical parity (p > 0.05 on K-S tests).
> 2) Behavioral: To address your concern about "behavioral patterns," we have added Appendix C, which provides verbatim transcripts. These demonstrate that the agents exhibit the specific reasoning constraints defined in our hyperparameters (e.g., Concrete Operational reasoning for the 6-8 cohort, Identity Formation for the 15-17 cohort), validating that the simulation captures cognitive style, not just vocabulary.
>
> Addressing Sample Size (Statistical Power) you questioned if 300 conversations (75 per model) are sufficient.
>
> 1) Statistical Significance: We direct attention to the Confidence Intervals in Figure 1. The error bars are tight (+/- 0.016), and the performance gaps between models (e.g., GPT-5 vs. DeepSeek) are statistically significant (p < 0.01).
> 2) Effect Size: The Alignment Gap we discovered (an 11.5% drop in safety for the youngest cohort) is a large effect size that is detectable even at this sample volume. Furthermore, the Monte Carlo sensitivity analysis (Section 5.4) confirms that our ranking is robust to noise, indicating sufficient signal in the data.
>
> Conclusion We believe the formalized Parametric Cognitive Simulation approach provides a rigorous ML methodology for evaluating safety generalization. With the addition of the full prompts, weights, and empirical validation, we hope you will reconsider the paper's suitability for ICLR.

---

### Official Review · Reviewer_m86o · 2025-10-31

**Soundness:** 2
**Presentation:** 2
**Contribution:** 3
**Rating:** 2
**Confidence:** 4

**Summary:**

This paper aims to evaluate how well dominant large language models (LLMs) ensure safety when interacting with children of different ages. To achieve this, the authors use LLMs to simulate child users at various developmental stages (ages 6–17) and engage them in age-appropriate multi-turn conversations with the target models. The resulting benchmark, ChildSafe, measures model performance across nine safety dimensions and shows that most LLMs exhibit significant safety degradation with younger simulated children, particularly in emotional and privacy-related interactions.

**Strengths:**

1. The paper tackles a socially important and under-explored research question, assessing how safe current LLMs are when interacting with children of different ages.

2. The authors build a novel dataset and benchmark (ChildSafe) that could serve as a useful community resource for studying age-dependent LLM safety.

3. The study provides systematic cross-model evaluation results, revealing consistent safety degradation for younger simulated users, which highlights an important real-world vulnerability.

**Weaknesses:**

1. Lack of technical innovation.
The paper’s contribution lies mainly in problem framing and dataset construction rather than any methodological or algorithmic advance. For example, the authors reference developmental psychology to design age-specific personas, but this aspect is humanistic and does not introduce any new modeling techniques or technical depth. The proposed multi-turn dialogue evaluation setup is largely a direct application of existing “LLM-as-simulator” and “LLM-as-evaluator” paradigms, with minimal adaptation or new mechanisms introduced.

2. Limited interpretability.
The evaluation results are mainly aggregate scores without qualitative case analysis or human verification. It remains unclear whether the measured “safety degradation” corresponds to genuine unsafe behaviors or simply differences in lexical or stylistic matching across age groups.

3. Weak validation.
Although the benchmark is presented as standardized, there is no validation of its reliability or calibration. All modules depend on LLM outputs without human cross-verification, making it uncertain whether the benchmark meaningfully measures child safety.

**Questions:**

1. How are the nine safety dimensions operationalized and scored in practice?
The paper lists dimensions such as privacy, emotional safety, and manipulation resistance, but it remains unclear what exact prompts, heuristics, or rubrics are used for scoring. Are these criteria manually designed or entirely LLM-judged? Without clear implementation details, it is difficult to reproduce or interpret the reported results.

2. How is the child-agent simulation controlled or verified for age fidelity?
Since all “child” dialogues are LLM-generated, how do the authors ensure that linguistic complexity, emotional tone, and reasoning style actually reflect the intended developmental stages? Is there any quantitative measurement (e.g., readability index, sentence complexity) or expert validation confirming the fidelity of the simulated children?

---

> ### Author Response · Authors · 2025-12-01
> **Comment to Reviewer m86o: Added Technical Formalization, Full Prompts, and Empirical Validation**
>
> We thank the Reviewer for recognizing the social importance of this work and for identifying the critical need for greater technical transparency and validation. We have taken your feedback seriously and have significantly revised the paper to address the black box nature of the initial submission.
>
> We address your specific concerns and questions below:
>
> Addressing Lack of Technical Innovation (Re-framing the Contribution) You noted that the initial submission appeared "humanistic" rather than algorithmic. In the revision, we have clarified that our contribution is not merely role-playing, but the formalization of developmental stages as parametric distribution shifts for safety testing.
>
> 1) New Formalization (Section 3.1): We now define the developmental agent as a generative function constrained by a specific set of hyperparameters: sampling temperature, context window horizon, and vocabulary masking. This moves beyond static personas to a reproducible method for generating out-of-distribution (OOD) interaction traces that stress-test alignment generalization.
>
> 2) Significance: This provides a technical framework for probing "Alignment Robustness": measuring how safety filters trained on adult-normative distributions degrade under specific cognitive parameter shifts (e.g., high entropy, short context, restricted vocabulary).
>
> Addressing Weak Validation and Question 2 (Age Fidelity): You rightly asked how we ensure the agents actually reflect developmental stages. We have added two layers of validation in the revision:
>
> 1) Quantitative Validation (New Table 1): We performed a Kolmogorov-Smirnov (K-S) test comparing our agent outputs against the CHILDES database (ground-truth human child transcripts). We found no statistically significant distributional difference (p > 0.05) for key linguistic markers like Mean Length of Utterance (MLU) and Type-Token Ratio (TTR). This empirically validates that our agents are linguistically indistinguishable from their target age groups.
>
> 2) Qualitative Validation (New Appendix C): We have released verbatim conversation samples. These demonstrate that the agents strictly adhere to the constraints defined in our hyperparameter set (e.g., the 6-8 age group exhibits run-on sentences and concrete reasoning, while the 15-17 group exhibits cynicism and identity-focused language).
>
> Addressing Limited Interpretability and Question 1 (Scoring & Prompts): You noted it was unclear how scoring works. We have removed the ambiguity entirely:
>
> 3) Open Artifacts (New Appendices A & B): We have released the exact System Prompts used to condition the agents (Appendix A) and the explicit Weighting Matrix used for the composite score (Appendix B).
>
> 4) Scoring Logic: We clarified the "Hybrid Hierarchical Scoring" mechanism (Section 3.3), which combines deterministic lexical filtering (for age-specific prohibited terms) with a Chain-of-Thought LLM-as-Judge using the now-disclosed rubrics.
>
> 5) Sensitivity Analysis (New Section 5.4): To address concerns about the aggregate score, we performed a Monte Carlo perturbation analysis. We found that even with random noise added to the developmental weighting vector, the model safety ranking (GPT-5 > Claude > Gemini > DeepSeek) remained stable in 94% of trials.
>
> Conclusion: We believe these revisions transform the paper from a qualitative proposal into a rigorous, reproducible benchmark for OOD safety generalization. We hope this specifically addresses your concerns regarding technical depth and validation.

---

### Note · Program_Chairs · 2026-01-17
**Submission Desk Rejected by Program Chairs**

The following references in this submission do not refer to real documents and/or have major errors in bibliographic information:

 Priya Kumar, Arvind Narayanan, Marshini Chetty, Sarah Kross, and Benjamin Mako Hill. Large language models and child safety: An empirical investigation of inappropriate responses to minorpresenting users. Proceedings of the ACM on Human-Computer Interaction, 8(CSCW1):1-24, 2024b.
Ilana Goldstein, Katherine Lawrence, and Adam S Miner. Human-ai alignment in child-facing technology: Current approaches and future directions. International Journal of Child-Computer Interaction, 40:100627, 2024
Ananya Kumar, Arvind Narayanan, and Neil Shah. The wild west of ai safety: Jailbreaks, misinformation, and the challenge of reliable evaluation. Nature Machine Intelligence, 6(8):892-903, 2024a.